# Radiomics analysis for the early diagnosis of common sexually transmitted infections and skin lesions

Jiajun Sun[1,2], Zhen Yu[3,4]*, Yingping Li[5], Janet M. Towns[1,2], Lin Zhang[6,7], Jason J. Ong[1,2], Zongyuan Ge[3,4], Christopher K. Fairley[1,2], Lei Zhang[1,2,8,9]*

1 Melbourne Sexual Health Centre, Alfred Health, Melbourne, Victoria, Australia, 2 School of Translational Medicine, Faculty of Medicine, Nursing and Health Sciences, Monash University, Clayton, Victoria, Australia, 3 AIM for Health Lab, Monash University, Clayton, Victoria, Australia, 4 Faculty of IT, Monash University, Clayton, Victoria, Australia, 5 School of Artificial Intelligence, Xidian University, Xi'an, China, 6 Suzhou Industrial Park Monash Research Institute of Science and Technology, Suzhou, China, 7 School of Public Health and Preventative Medicine, School of Medicine, Nursing and Health Sciences, Monash University, Clayton, Victoria, Australia, 8 Phase I Clinical Trial Research Ward, The Second Affiliated Hospital of Xi'an Jiaotong University, Xi'an, China, 9 China-Australia Joint Research Center for Infectious Diseases, School of Public Health, Xi'an Jiaotong University Health Science Center, Xi'an, China

* Zhen.Yu@monash.edu (ZU); lei.zhang1@monash.edu (LZ)

## Abstract

Early identification of sexually transmitted infection (STI) symptoms can prevent subsequent complications and improve STI control. We analysed 597 images from STIAtlas and categorised the images into four typical STIs and two skin lesions by the anatomical sites of infections. We first applied nine image filters and 11 machine-learning image classifiers to the images. We then extracted radiomics features from the filtered images and trained them with 99 models that combined image filters and classifiers. Model performance was evaluated by area under curve (AUC) and permutation importance. When the information of infection sites was unspecified, a combined Gradient-Boosted Decision Trees (GBDT) classifier and Laplacian of Gaussian (LoG) filter model achieved the best overall performance with an average AUC of 0.681 (95% CI 0.628-0.734). This model predicted best for lichen sclerosus (AUC = 0.768, 0.740-0.796). The incorporation of infection site information led to a substantial improvement in the model's performance, with 22.3% improvement for anal infections (AUC = 0.833, 0.687-0.979) and 3.8% for skin infections (AUC = 0.707, 0.608-0.806). Lesion texture and statistical radiomics features were the most predictive for STIs. Combining machine learning and radiomics techniques is an effective method to categorise skin lesions associated with STIs clinically.

### Author summary

We developed an artificial intelligence tool that can help identify sexually transmitted infections (STIs) from photographs of skin lesions. Using machine

which permits unrestricted use, distribution, and reproduction in any medium, provided the original author and source are credited.

**Data availability statement:** All data in this study is public available on https://stiatlas.org/.

**Funding:** The author(s) received no specific funding for this work.

**Competing interests:** The authors have declared that no competing interests exist.

learning and a technique called radiomics—which extracts detailed information about texture and shape from medical images—we analysed 597 images from the STI Atlas database covering four common STIs and two skin conditions. Our approach combines computer algorithms with radiomics to automatically detect features in skin images that might indicate specific infections. We found that when we included information about where on the body the infection appeared (genitals, anus, or other skin areas), our tool's accuracy improved significantly. The biggest improvement was for anal infections, where accuracy increased by over 22%. This technology could be particularly valuable in areas with limited access to healthcare specialists, allowing people to take photographs with their smartphones for preliminary assessment. While not intended to replace clinical diagnosis, our tool could help people decide whether they need urgent medical attention. This aligns with global health efforts to improve early detection and treatment of sexually transmitted infections, potentially reducing transmission and complications in communities worldwide.

## Introduction

Sexually transmitted infections (STIs) pose a major public health challenge. According to the World Health Organization (WHO), an estimated 374 million new infections with trichomoniasis, chlamydia, gonorrhoea, and syphilis occurred globally in 2020 [1]. Furthermore, the prevalence of STIs has been steadily increasing in recent years. For example, there was a 72.5% increase in the number of trichomoniasis cases between 1990 and 2019 which rose from 205.4 million to 354.5 million. Similarly, syphilis cases rose by 60% to 14.1 million from 8.8 million during the same period [2]. In 2020, out of 7.1 million new syphilis cases, 1 million occurred among pregnant women aged 15–49 years old [3]. Pregnant women living with HIV and with concomitant STIs face a twofold higher risk of preterm delivery [4], and untreated syphilis during pregnancy carries a 25% risk of stillbirth. Furthermore, the vertical transmission rate of untreated syphilis from mother to child in the third trimester of pregnancy can be as high as 60–100% [5]. In addressing the concerning rise of STIs, the WHO has set ambitious targets to promote the end of STIs by 2030 [6].

To achieve the goal of ending the spread of STIs, the WHO has developed specific strategies for the prevention and early detection of STIs. The WHO strategies on HIV and sexually transmitted infections [6] all highlight the importance of early diagnosis for reducing transmission and the impact of STIs on individuals and communities. Specifically, the WHO required that by 2030, the annual coverage of syphilis and gonorrhoea screening among key populations exceeds 90%. The WHO advocates for the implementation of targeted strategies aimed at early detection and treatment, emphasising the importance of timely interventions to prevent complications and transmission. The WHO advocates for the implementation of the "foster innovations for impact" strategy, prioritizing the combination of the latest technologies to establish screening tools for the early detection and treatment of STIs.

The current model for STI care where individuals attend a health care service depends on them accurately recognising symptoms of STIs and there being adequate and accessible health services [7–9]. To overcome these challenges, recent research has explored the potential of artificial intelligence (AI) as a tool assisting individuals in determining if they need urgent STI care or not thereby facilitating urgent presentations for those with STIs and avoiding unnecessary presentations for those without STIs [10]. Machine learning methods consistently outperform traditional multivariate logistic regression in the prediction of infection risk for HIV/STIs based on clinical records [11]. MySTIRisk uses self-reported information from individuals attending an STI clinic in an AI-based risk assessment tool capable of predicting syphilis with an AUC of 0.84 [12,13]. Perhaps the greatest role for AI is in Low- or Middle-Income Countries (LMICs), where trained clinicians and medical resources are scarce, and traditional healthcare infrastructure is limited [14,15]. The introduction of AI-assisted tools could revolutionise STI care by bringing it directly to individuals' mobile phones.

A further tool that could potentially improve the accuracy of AI for STI diagnosis is radiomics [16]. AI using radiomics was initially developed for tumour detection, where it demonstrated high accuracy. However, AI tools are often perceived as black boxes, lacking clinically persuasive interpretability [17]. But radiomics is less of a black box, radiomics extracts clinically significant information, such as texture and shape details, from medical images [18]. The advantages of radiomics tools have meant that they have been rapidly applied to medical image processing such as computed tomography (CT), magnetic resonance imaging (MRI) and positron emission tomography (PET) to facilitate a more accurate diagnosis [19]. However, we believe the application of radiomics for STI detection is a relatively unexplored area.

The other factor that could potentially improve the AI diagnosis of STIs is the anatomical site of the lesion. As far as we know, none of the existing studies investigate STI lesions in multiple sites. We hypothesise that the inclusion of infection site information in the models may substantially improve the performance of the model. This variability of images at different anatomical locations may be a key factor that influences the radiomics-based AI models for STI detection. For instance, early syphilis infection presents distinct clinical manifestations depending on the body area affected [20]. On the trunk and limbs, it typically appears as a macular or papulosquamous eruption, characterised by flat or slightly raised lesions. In contrast, within the genito-anal region, the infection often manifests as confluent nodules of condyloma latum, which are larger, raised lesions that tend to merge together [21]. In research on STI skin diagnosis tools, this factor has not been adequately investigated or analysed.

We aim to develop an AI-assisted testing tool for common STIs by integrating machine learning and radiomics approaches. The Melbourne Sexual Health Centre (MSHC), a public sexual health clinic in Melbourne, Australia, has collected a substantial database of images, and it has been publicly available at STI Atlas [22]. This exploratory study may facilitate earlier detection of several types of STIs which could lead to skin lesions, aiding in the prevention of further transmission and contributing significantly to public health efforts.

## Methods

We developed models for the diagnosis of skin lesions from four common sexually transmitted infections and two skin lesions. We built these models by combining radiomics technology and deep learning classifiers. Before building models, we collected STI images from STI ATLAS (https://stiatlas.org/), which supplied skin images with infection types and body sites. All diagnoses for the cases in the STI Atlas dataset have been confirmed with serological and/or molecular testing, serving as the gold standard for STI diagnosis. To ensure the reliability of manual segmentation, two STI specialists independently segmented these images, with inter-observer agreement evaluated by a third research member. Additionally, one specialist re-evaluated the same subset after a one-month interval to assess intra-observer consistency. In cases of discrepancies, the specialists discussed and reached a consensus to finalise the segmentation. Then, we used the radiomics technique to extract radiomics features and selected low-correlation features based on the Pearson correlation coefficient. Finally, we trained and compared several deep-learning classifiers. The structure of our model is shown in S1 Fig. The models we developed could identify common STIs and can be applied to self-diagnosis.

PLOS Digital Health

**STIs images collection**

We trained our model on STI ATLAS. It provides royalty-free, high-quality images of STIs as an educational resource. We extracted 945 of the images and infection site information on the website and saved it as an original dataset. JT, JO and CKF are STI specialists with extensive clinical experience. Before proceeding, JO and CKF each conducted a thorough review of all images to confirm both the presence of an STI and the absence of identifying information. Images with different opinions were removed. Following the initial review, CKF, JO, and LZ collaborated to re-evaluate the remaining images, summarising the depicted body sites and types of sexually transmitted infections. This resulted in the identification of 21 distinct infection types across seven different body sites. Our initial dataset included images of diverse STIs, but some were too uncommon (less than 10 cases each). After excluding these, we had a final dataset of 597 images for analysis.

**Manual lesion segmentation for the region of interest**

Regions of interest (ROIs) refer to manually labelled polygonal regions on the image. Radiomics techniques could extract extra radiomics features from each ROI. The accuracy of the labelled regions has a crucial impact on the model accuracy. We utilised the LabelMe tool [23] for ROI labelling. To ensure consistency and reliability, JJS initially trained JT and CKF on the use of the LabelMe tool, emphasising the importance of expanding segmentation areas beyond the immediate infection to capture subtle changes like colour variations. JT and CKF then independently completed the manual segmentation, with JJS overseeing quality control, documenting progress, and harmonising any discrepancies through regular meetings.

**Images derived by filters**

We applied nine built-in filters provided by pyradiomics [24], an open-source Python package for the extraction of radiomics features from medical imaging, to generate derived images from the original skin images. Applying image filters significantly enhanced key features in the original images, particularly texture and boundary details, allowing for clearer differentiation between infected and normal areas. These image filters included the original grey filter, gaussian laplace (LoG) filter, gradient filter, square filter, square root filter, logarithm filter, exponential filter, two-dimensional local binary pattern (LBP2D) and wavelet filters targeting specific edge features (four wavelet sub-filters). We applied each of these nine filters to the original images, generating the derived images shown in S2 Fig. These filtered images then served as inputs for radiomics feature extraction.

**Radiomics feature extraction and selection**

From each ROI in the derived images, we extracted 102 radiomics features using the Python package pyradiomics. To comprehensively analyse the infection areas, we categorised the extracted 102 radiomics features into four groups. The first group encompassed statistical features, incorporating 18 first-order features such as the range of grey values. The second group focused on shape features, including nine 2D shape-based (shape2D) features. The third group involved texture features, comprising 24 grayscale co-occurrence matrix (GLCM) features, 16 grayscale run length matrix (GLRLM) features, 16 grayscale size region matrix (GLSZM) features, and five neighbouring grey tone difference matrix (NGTDM) features. The final group centred on voxel dependence features, including 14 grayscale dependence matrix (GLDM) features. Through meticulous comparisons of filter performance, we pinpointed the most effective filter for each prediction classifier.

We filtered radiomics features based on their correlation to ensure robust and accurate analysis. Using the Pearson correlation coefficient, we identified and eliminated features exhibiting strong dependencies with others. We considered features with absolute correlation values exceeding 0.8 as highly correlated and likely to introduce redundancy or

instability into prediction models. Consequently, we initially identified pairs of such highly correlated features and then calculated the sum of the absolute correlation coefficients between each of these pairs and all other remaining features. Subsequently, we removed the feature from each correlated pair with the highest sum of correlations with the other features. By removing strong correlation features, we mitigated their potential adverse effects on prediction accuracy and ensured the model's focus on independent, informative features.

We standardised the radiomics features prior to inputting them into the machine learning models. This was crucial to guarantee unbiased training and to ensure each feature contributed optimally. The raw values of these features varied significantly, a factor that could potentially bias the model's learning process. Features with broader value ranges might overshadow those with smaller ranges during training, even though the latter could be equally or more informative. To rectify this, we employed the standard scaling method, transforming each feature to have a mean of 0 and a variance of 1. This method ensures that no single feature dominates or gets ignored due to its original scale. Every radiomics feature contributed meaningfully to the model's learning and prediction, resulting in more reliable and unbiased results.

## Classification model development and interpretation

We conducted a comprehensive evaluation using 11 widely used machine learning classifiers from the PyTorch library [25]. Each classifier was trained on nine image filter-derived images, resulting in a total of 99 models. We further refined the dataset by splitting it based on the infection site, creating dedicated models for each site. Given the limited number of infection cases, we opted for a 5-fold cross-validation strategy for robust performance assessment (S3 Fig). This involved dividing the images into five folds and sequentially training the classifiers on four folds while testing on the remaining fold. To address the class imbalance in our dataset and ensure robust model evaluation, we employed Stratified ShuffleSplit cross-validator for cross-validation. This approach maintains the percentage of samples for each class across the splits while allowing for random sampling with replacement. Due to the stratified nature of the sampling and the need to maintain balanced class proportions, some samples may be selected multiple times across different folds. Each iteration yielded an AUC score, which was then averaged to provide a more reliable performance metric. True/false positive and negative results were further visualised using confusion matrix diagrams.

We used the permutation importance method to evaluate the significance of radiomics features on top models and understand how it makes its predictions. The permutation importance method measured the radiomics feature's importance by shuffling its values across all cases and observing the change in model performance. We repeated the shuffling process 100 times for each radiomics feature and calculated the average decrease in AUC. Higher drops indicated a greater influence on the model's performance, revealing the most critical features for its predictions. This analysis revealed the most influential radiomics features in the models, providing valuable insights into its decision-making process and ultimately boosting understanding of the model's interpretability.

## Ethics statement

Permission to conduct the project was submitted and approved by the Ethics Committee of the Alfred Hospital (Ethics Project No. 191/23). The images in this study are from STI Atlas and do not contain any identifiable information. This study does not include factors necessitating patient consent.

## Results

### Description of image data

We focused on four STIs and two skin lesions across three key body sites: genitals, anus, and other skin areas. "Genitals" referred to external genitalia, including the penis, scrotum, vulva, and perineum. "Anus" specifically denoted the perianal region. "Other skin" referred to areas excluding the genitals and anus, covering other body regions. These prevalent

conditions captured in our dataset included sexually transmitted infections that caused skin lesions, such as early syphilis (143 cases), herpes (128 cases), warts (127 cases), and molluscum contagiosum (66 cases), as well as general skin lesions, including lichen sclerosus (145 cases) and tinea (61 cases). S1 Table provides a description of our dataset, detailing the distribution of infections across the sites with herpes being documented in 36 genital and 25 other skin cases, lichen sclerosus in 127 genital and 18 other skin cases, molluscum contagiosum in 12 genital and 54 other skin cases, early syphilis in 45 genital, 12 anal, and 86 other skin cases, tinea exclusively in 61 skin cases, and warts in 54 genital, 15 anal, and 52 other skin cases.

We categorized the images into three groups based on the infection site: genitals (274 images), skin (296 images), and anus (26 images), as detailed in S1 Table. The performance of the models for unspecified infection sites is detailed in S2–S5 Tables. S6–S9 Tables presents a comprehensive performance comparison of all models across the three infection sites. The confusion matrix in Fig 1 represents the aggregated results across all cross-validation folds, rather than predictions on unique samples. This methodology was chosen to provide a comprehensive view of the model's performance across different data splits while maintaining class balance. Some samples may appear multiple times in the final confusion matrix, but this stratified sampling strategy is essential for reliable performance evaluation when dealing with imbalanced datasets.

## Models performance when infection sites were unspecified

Fig 2A demonstrated the model performance for four STIs and two skin lesions when infection sites were unspecified. The combined GBDT with LoG filter performed best, with an average AUC of 0.681 (95% CI, 0.628-0.734) on the test dataset.

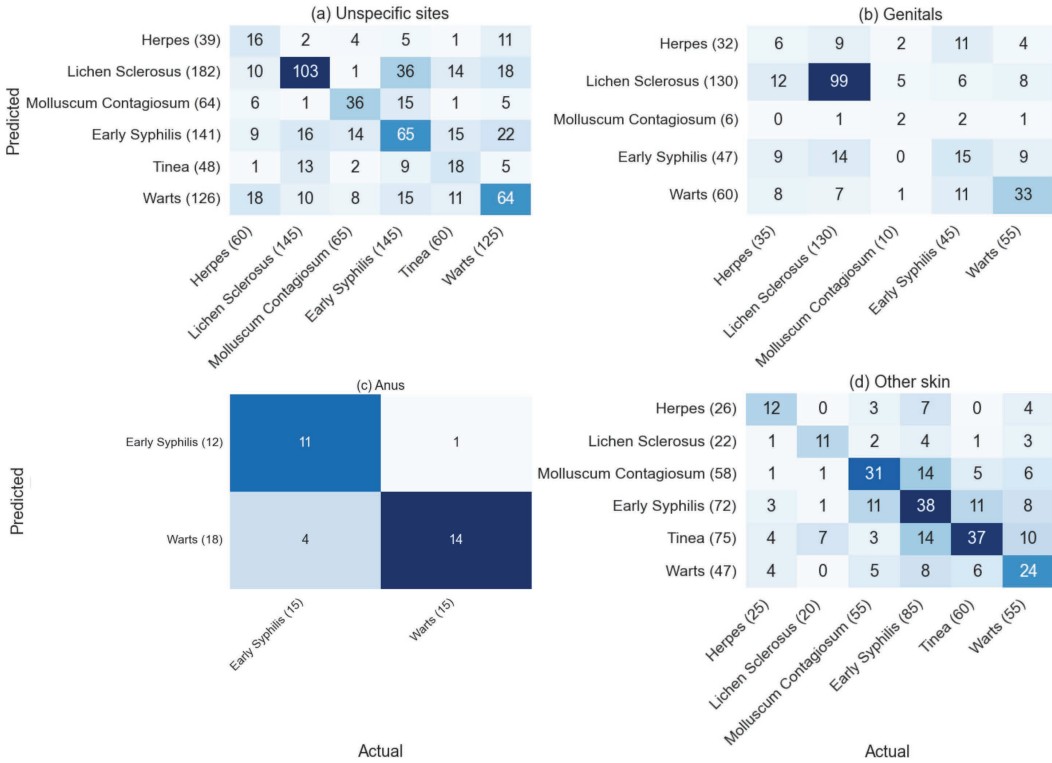

**Fig 1. Confusion matrices for the models predicted results on the testing dataset on (a) unspecific sites; (b) genitals; (c) anus; (d) other skin.**

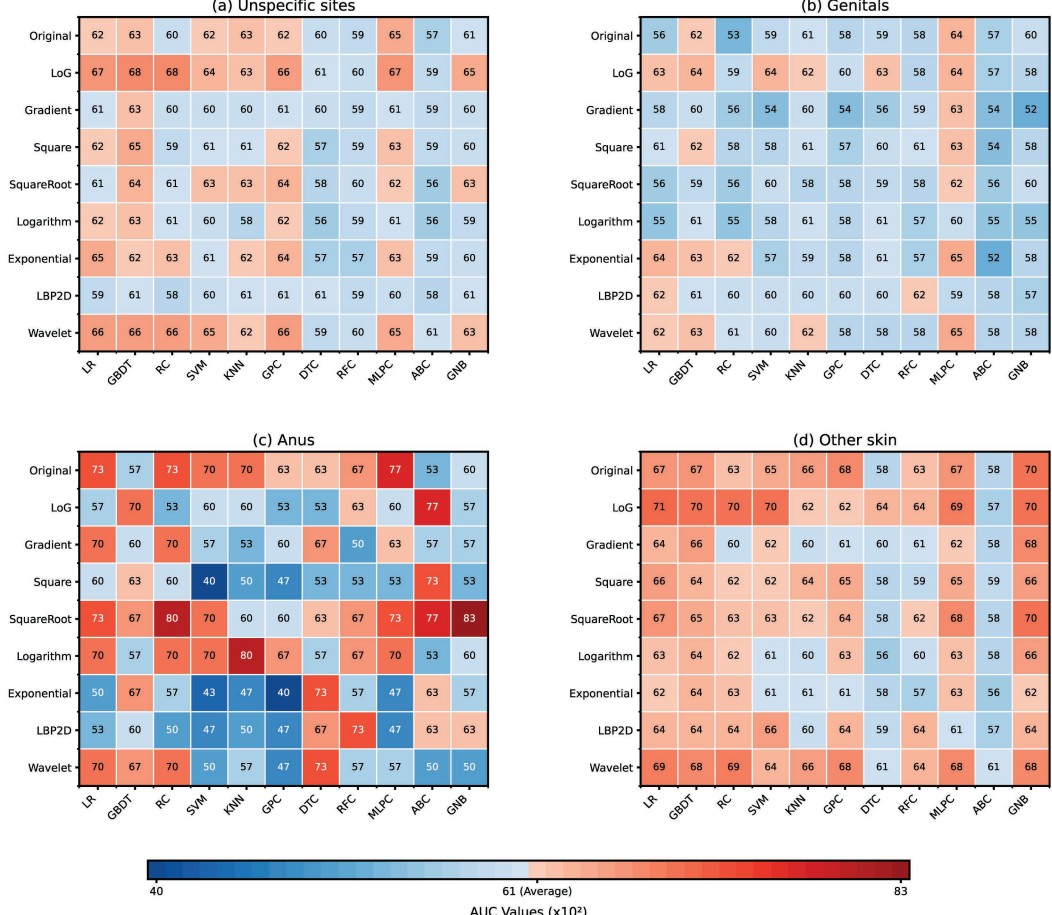

**Fig 2. Models' AUC (×10²) with nine images filter and eleven classifiers on (a) unspecific sites; (b) genitals; (c) anus; (d) other skin.** *ABC: Adaptive Boosting Classifier, DTC: Decision Tree Classifier, GBDT: Gradient Boosted Decision Trees, GNB: Gaussian Naive Bayes, GPC: Gaussian Process Classifier, KNN: K-nearest Neighbors, LR: Logistic Regression, MLPC: Multi-layer Perceptron Classifier, RC: RidgeClassifier, RFC: Random Forest Classifier, SVM: Support Vector Machine.

With this best-performed model shown in Fig 3, the prediction of lichen sclerosus showed the highest AUC of 0.768 (95% CI, 0.74-0.796). Followed by molluscum contagiosum, with an AUC of 0.751 (95% CI, 0.701-0.801). Warts ranked third, with an AUC of 0.691 (95% CI, 0.656-0.726).

The confusion matrix for the highest AUC models when infection sites were unspecified is shown in Fig 1A. The model correctly identified 16 out of 60 cases (26.7%) with herpes, 103 out of 145 cases (71.0%) with lichen sclerosus, 36 out of 65 cases (55.4%) with molluscum contagiosum, 65 out of 145 cases (44.8%) with early syphilis, 18 out of 60 cases (30.0%) with tinea, and 64 out of 125 cases (51.2%) with warts. Overall, the accuracy of the model with unspecific sites is 50.3%.

The top 10 radiomics features, as determined by permutation importance for the leading models in Fig 4A, were ranked from most to least significant. Maximum was the most important feature for the unspecific sites model. Randomly disrupting it caused the model AUC to decrease by 0.023 (95% CI 0.022-0.024). And the majority of the top 10 radiomics features are statistical features and textural features. This suggested that the best model was primarily relying on the statistics of pixels and texture of the lesions to make predictions.

## Models performance on genital conditions

As shown in Fig 2B, for infections on the genitals, the combined MLP classifier with the exponential filter showed the best performance, with an average AUC of 0.647 (95% CI, 0.553-0.741) on the test dataset. As shown in Fig 3, the genitals model was slightly more effective at predicting lichen sclerosus (AUC, 0.774, 0.78% improved) and warts (AUC, 0.739, 6.95% improved) than when no specific infection site was considered. As shown in Fig 4B, the major axis length was the most important feature for the genitals model. Randomly disrupting it caused the model AUC to decrease by 0.075 (95%CI 0.07-0.08).

The confusion matrix with genitals site information in 1B showed that the model correctly identified 6 out of 35 cases (17.1%) with herpes, 99 out of 130 cases (76.1%) with lichen sclerosus, 2 out of 10 cases (20.0%) with molluscum contagiosum, 15 out of 45 cases (33.3%) with early syphilis, and 33 out of 55 cases (60.0%) with warts. Overall, the accuracy of the model with genitals site information is 56.4%.

## Models performance on anus infections

Regarding infections on the anus shown in Fig 2C, the combined Gaussian Naive Bayes with the square root filter was the most effective, with an average AUC of 0.833 (95% CI, 0.687 - 0.979). Compared with models without infected body site specification in Fig 3, the AUC made a substantial increase of 0.152 (22.3%), reflecting a notable improvement in the model's predictive accuracy for early syphilis and warts infections. As shown in Fig 4C, cluster shade was the most important feature of the anus model. Randomly disrupting it caused the model AUC to decrease by 0.098 (95% CI 0.084-0.11).

The confusion matrix in Fig 1C with anus site information showed that the model correctly identified 11 out of 15 cases (73.3%) with early syphilis, and 14 out of 15 cases (93.3%) with warts. Overall, the accuracy of the model with anus site information is 83.3%.

## Models performance on other skin infections

In the case of infections on the skin shown in Fig 2D, the combined Logistic Regression with the LoG filter demonstrated superior performance, with an average AUC of 0.707 (95% CI, 0.608-0.806). When compared with models that did not specify infected body sites in Fig 3, the AUC increased by 0.026 (3.82%), indicating a significant enhancement in the

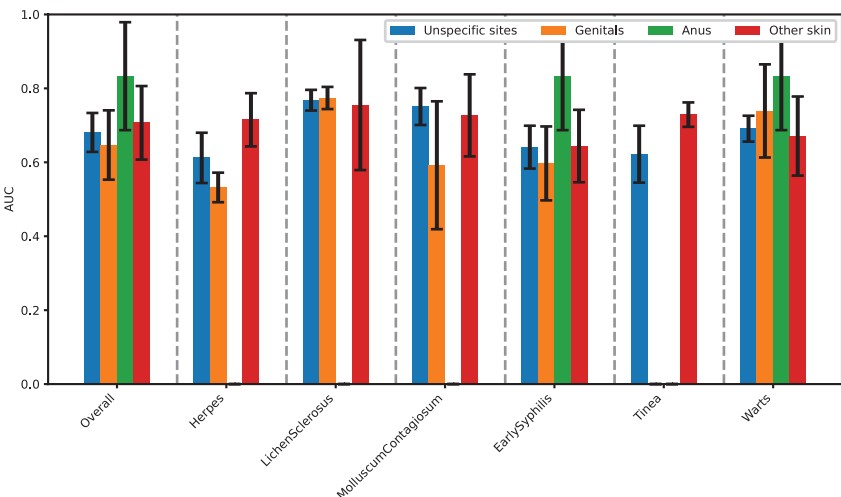

**Fig 3. Comparison of the AUC of the best infection prediction model for infections on unspecific sites, genitals, anus and other skin.**

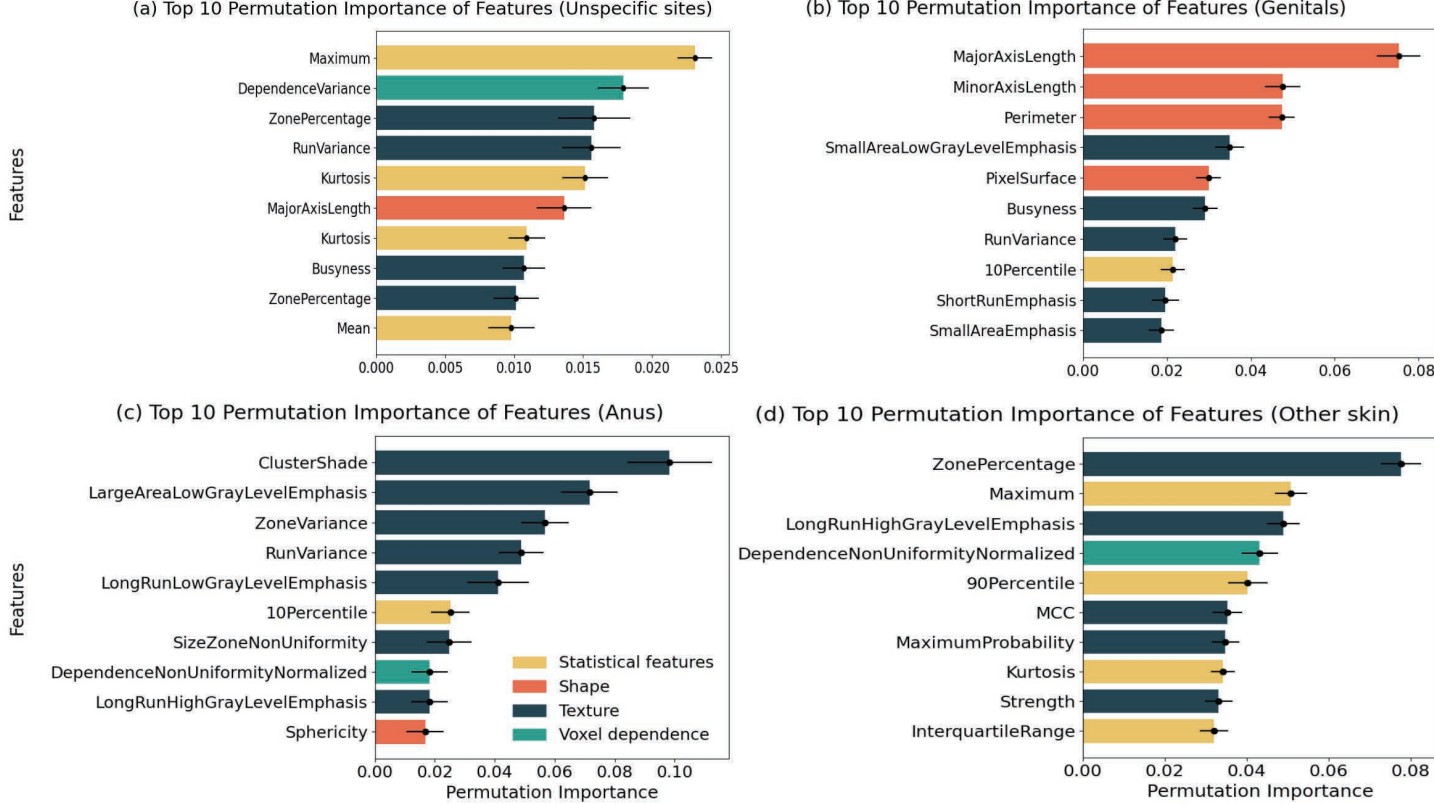

**Fig 4. The top rank 10 radiomics features in the best models using a permutation importance analysis.**

model's ability to predict skin infections accurately. As shown in Fig 4D, zone percentage was the most important feature for skin models. Randomly disrupting it caused the model AUC to decrease by 0.078 (95%CI 0.073-0.083).

The confusion matrix with skin in Fig 1D showed that the model correctly identified 12 out of 25 cases (48.0%) with herpes, 11 out of 20 cases (55.0%) with lichen sclerosus, 31 out of 55 cases (56.4%) with molluscum contagiosum, 38 out of 85 cases (44.7%) with early syphilis, 37 out of 60 cases (61.7%) with tinea, and 24 out of 55 cases (43.6%) with warts. Overall, the accuracy of the model with skin site information is 51.0%.

### Models comparison and radiomics feature interpretability

Fig 3 illustrates an enhancement in the model's predictive accuracy for most infections when the site of infection is taken into account. The prediction of early syphilis at the anus site showed the most substantial improvement. In terms of feature importance in Fig 4, texture features ranked highest, followed by statistical features. For two out of the four infection sites, texture features emerged as the most crucial, while for the genitals site, shape features were predominant. Additionally, statistical features were identified as the second most significant at three sites. This pattern suggests that the radiomics tool predominantly relies on the texture of the STI skin lesions for making predictions.

### Discussion

Our study integrated machine learning and radiomics approaches to identify key features of STIs and skin lesions based on a sizable database of skin images. The ridge classifier, when paired with a LoG filter, showed superior performance,

especially for non-specific sites. Integrating information about the infection site into the model markedly enhanced its accuracy, with notable improvements for anal infections. The AUC for the prediction of early syphilis infection in the peri-anal region increased significantly by 37.5% (AUC improved from 0.606 to 0.833). To our knowledge, this is the first study employing radiomics for STI diagnosis from skin images. We employed permutation importance to pinpoint texture and statistical features are very important in identifying STIs in all infection sites.

Our findings underscore the potential of radiomics technology in self-testing for sexually transmitted infections. We propose an innovative approach that synergizes machine learning with radiomics, fostering an AI-driven, convenient self-testing method using skin images. This strategy significantly boosts accuracy and efficiency, bypassing invasive procedures and facilitating accessible testing via smart devices. This is particularly beneficial for remote or underserved communities with limited access to specialists. Enhancing the image dataset and integrating clinical data could further refine these AI self-diagnostic tools, increasing their precision. Compared to conventional healthcare interactions, our tool offers digital healthcare solutions that support private, anonymous, and secure early diagnosis. This advancement aligns with timely and effective sexual health services, contributing to the WHO's global STI control efforts [1].

Our study found that including infection sites as the models' parameter could help to improve the classifier accuracy when predicting STIs. By incorporating body site information into the model, we were able to achieve a significant improvement in accuracy. This finding has important implications for the development of clinical decision support systems for STI diagnosis. These results suggest that considering body site information can significantly improve the accuracy of the model for predicting STIs. Specifically, the model's accuracy for predicting lichen sclerosus, early syphilis and warts increased significantly when body site information was considered. The most significant increase in predictive performance was observed for early syphilis, with an average increase in AUC of 30.0%. This is likely because body site information can provide additional clues about the type of STI that is present. For example, lichen sclerosus primarily affects the genital area and around the anus [26,27], early syphilis often leads to lesions on skin [28], while warts are more likely to occur on the anus [29]. Other clinical data such as information on symptoms presentation and sexual behavioural patterns of the individuals may also improve the model's accuracy [30,31].

Our study found the texture and statistical features emerged as highly sensitive to skin lesions, demonstrating a positive correlation with accurate STI classification. The predictive model on the genitals site emphasizes the importance of shape characteristics, revealing that different STIs occurring at the site exhibit unique shape characteristics. This suggests they hold valuable information for discriminating between STI and non-STI cases. The high importance of these features in our machine learning models implies they capture crucial differences in the texture or structure of medical images, potentially reflecting underlying STI pathology. Analysing feature importance within these models can therefore offer reasonable explanations for the skin-related symptoms observed in STIs. Ultimately, high-importance radiomics features shed light on the morphological characteristics of STI sites, providing valuable insights for both diagnosis and understanding of disease progression.

Our study also has several limitations. Firstly, the types of infections covered in our dataset are limited. We only focused on a small range of infection types commonly captured in our clinic. Secondly, there was an imbalance in the number of images representing each infection type within our dataset. We captured as many skin images as we could, but infections in some body sites are relatively rare. This imbalance contributed to the discrepancy between sensitivity and specificity in our model. Moreover, within each infection category, the available image count is constrained, potentially affecting the model's ability to discern subtle nuances associated with specific infections. Furthermore, the accuracy of our models is contingent upon the manual annotation of infection regions. Relying on hand-annotated data introduces a degree of subjectivity and potential bias, which may influence the model's performance, mainly when applied to diverse and real-world clinical scenarios. In our forthcoming efforts, we aim to address this limitation by delving into the development of automated annotation methods for infection sites. By exploring techniques for unbiased and high-precision region annotation, we anticipate enhancing the accuracy and reliability of our models.

In conclusion, our study demonstrates that combining machine learning with radiomics can effectively utilise skin images for early STI diagnosis. Incorporating infected body site information into models significantly improves predictive accuracy.

## Supporting information

**S1 Fig. The structure of the typical STI prediction models.**
(DOCX)

**S2 Fig. Nine different filters.** (A) Original grey filter, (B) Gaussian Laplace (LoG) filter, (C) gradient filter, (D) square filter, (E) square root filter, (F) logarithm filter, (G) exponential filter,(H) two dimension local binary pattern (LBP2D), (I) HH Wavelet filter (extract diagonal features), (J) HL Wavelet filter (extract vertical features), (K) LH Wavelet filter (extract horizontal features), (L) LL Wavelet filter (the approximate image).
(DOCX)

**S3 Fig. Methods of dividing images into training and testing data.**
(DOCX)

**S1 Table. General information about the STIs images dataset.**
(DOCX)

**S2 Table. AUC results for ten classifiers with nine filters on unspecific sites.**
(DOCX)

**S3 Table. Accuracy results for ten classifiers with nine filters on unspecific sites.**
(DOCX)

**S4 Table. Specificity results for ten classifiers with nine filters on unspecific sites.**
(DOCX)

**S5 Table. Sensitivity results for ten classifiers with nine filters on unspecific sites.**
(DOCX)

**S6 Table. AUC results of the classifiers with three infection body sites.**
(DOCX)

**S7 Table. Accuracy results of the classifiers with three infection body sites.**
(DOCX)

**S8 Table. Specificity results of the classifiers with three infection body sites.**
(DOCX)

**S9 Table. Sensitivity results of the classifiers with three infection body sites.**
(DOCX)

## Author contributions

**Data curation:** Jiajun Sun.

**Formal analysis:** Jiajun Sun.

**Funding acquisition:** Jason J. Ong, Christopher K. Fairley.

**Investigation:** Jiajun Sun, Janet M. Towns, Lei Zhang.

**Methodology:** Zhen Yu, Yingping Li, Lei Zhang.

**Project administration:** Jiajun Sun.

**Resources:** Jason J. Ong, Christopher K. Fairley.

**Software:** Jiajun Sun.

**Supervision:** Jiajun Sun, Lin Zhang, Jason J. Ong, Zongyuan Ge, Christopher K. Fairley, Lei Zhang.

**Validation:** Jiajun Sun, Yingping Li.

**Visualization:** Jiajun Sun, Zhen Yu, Yingping Li, Lei Zhang.

**Writing – original draft:** Jiajun Sun, Yingping Li.

**Writing – review & editing:** Jiajun Sun, Zhen Yu, Janet M. Towns, Lin Zhang, Jason J. Ong, Zongyuan Ge, Christopher K. Fairley, Lei Zhang.

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
