## [Decision Letter · Decision Letter 0]

PDIG-D-24-00497Radiomics Analysis for the Early Diagnosis of Common Sexually Transmitted Infections and Skin LesionsPLOS Digital Health Dear Dr. Sun, Thank you for submitting your manuscript to PLOS Digital Health. After careful consideration, we feel that it has merit but does not fully meet PLOS Digital Health's publication criteria as it currently stands. Therefore, we invite you to submit a revised version of the manuscript that addresses the points raised during the review process. Please submit your revised manuscript within 60 days May 12 2025 11:59PM. If you will need more time than this to complete your revisions, please reply to this message or contact the journal office at digitalhealth@plos.org. Please include the following items when submitting your revised manuscript:* A rebuttal letter that responds to each point raised by the editor and reviewer(s). You should upload this letter as a separate file labeled 'Response to Reviewers '. This file does not need to include responses to any formatting updates and technical items listed in the 'Journal Requirements' section below.* A marked-up copy of your manuscript that highlights changes made to the original version. You should upload this as a separate file labeled 'Revised Manuscript with Track Changes '.* An unmarked version of your revised paper without tracked changes. You should upload this as a separate file labeled 'Manuscript '. If you would like to make changes to your financial disclosure, competing interests statement, or data availability statement, please make these updates within the submission form at the time of resubmission. Guidelines for resubmitting your figure files are available below the reviewer comments at the end of this letter. We look forward to receiving your revised manuscript. Kind regards, Hisham Al-Obaidi, PHDAcademic EditorPLOS Digital Health Hisham Al-ObaidiAcademic EditorPLOS Digital Health Leo Anthony CeliEditor-in-ChiefPLOS Digital Healthorcid.org/0000-0001-6712-6626 **Journal Requirements:**

 **Additional Editor Comments (if provided):****Reviewers' Comments:** Reviewer's Responses to Questions

**Comments to the Author**

1. Does this manuscript meet PLOS Digital Health’s publication criteria ? Is the manuscript technically sound, and do the data support the conclusions? The manuscript must describe methodologically and ethically rigorous research with conclusions that are appropriately drawn based on the data presented.

Reviewer #1: Partly

Reviewer #2: Partly

2. Has the statistical analysis been performed appropriately and rigorously?

Reviewer #1: Yes

Reviewer #2: Yes

3. Have the authors made all data underlying the findings in their manuscript fully available (please refer to the Data Availability Statement at the start of the manuscript PDF file)?

Reviewer #1: Yes

Reviewer #2: Yes

4. Is the manuscript presented in an intelligible fashion and written in standard English?

Reviewer #1: Yes

Reviewer #2: Yes

5. Review Comments to the Author

Reviewer #1: Comment: Radiomics Analysis for the Early Diagnosis of Common Sexually Transmitted Infections and Skin Lesions

1. Line 14-126 The authors asked specialists with extensive clinical experience to confirm the presence of an STI. It seems there is a lack of golden standard to determine STI. It is not sure how reliable about the subjective clinical judgment. The authors need to address this.

2. Line 114-115. The authors used two STI specialists to make manual segmentation to label the infection area. Is there any inter-observer and intra-observer studies to validate the reliability of the segmentation method? If there is any discrepancies among the specialists, what will be done to make the final decision?

3. The study needs to use accuracy, specificity and sensitivity to illustrate the results. I am afraid that since there are imbalance data in each category of skin lesion, this will lead to result with high sensitivity and low specificity. The authors should consider this.

Reviewer #2: The use of radiomics analysis is fairly new in medicine and apart from use in radiology and oncology, it's use is novice in genital dermatology and sexual health medicine. This is another 'tool' (Line 81) which used in conjunction with self-reported risk (presumably symptoms) of STI (Line 78-79) would allow an individual recognising 'signs' and directing them to health-services for confirmation through laboratory techniques and lead to a 'cure'. This would enhance a better 'self diagnosis' that individuals may use "Dr Google" for. It would also enhance a working diagnosis (or differential) of many primary care physicians and nurses, who may not have encountered genital ulcer infections such as syphilis, as the incidence remains low in Australia.

Line 111

4 common STIs and 2 skin lesions - the lesions of concern are reported in 210-212 and perhaps this could have been made clearer in the Methods. The only common STI discussed in Line 48-49 that seems relevant in this study, is syphilis; whereas the author(s) states that the mots global prevalent STIs are trichomiasis, gonococcal and chlamydial infection. The title is mis-leading in that the study looked at infections of genital lesion / ulcers / genital dermatosis.

Line 136

LabelMe tool - there is no reference as to what this 'tool' is - was it developed as part of this study; and if so, has it been validated / standardised. Need more information / reference to this 'tool'

Line 97 - 101 and 323

Has syphilis been diagnostically 'misclassified' as 'secondary syphilis'?

Was radiomics analysis able to define between a primary syphilitic chancre (particularly in the genital / anus) to secondary manifestations (such as Condylomata lata, mucous patches) with / without a 'skin' rash? If radiomics analysis was unable to distinguish these, I suggest using the term "Early Syphilis" versus "Secondary syphilis"

Line 214 / 218 - site of photograph; Table S1, S2, S3

The sites of the photographs need better clarity - what sites are 'genitals' ? (name sites) as well as "Skin" - e.g. keratinised skin versus 'non-keratinised"

References that need attention:

Line 366 and 377

Reference 1 and Reference 6 need to be clearly written per acaedemic standards. Was the source the World Health Organisation (WHO); and it would be preferably followed by a link to a website and when this was last accessed.

Line 420

Reference 22 - this reference is not sufficiently provided as is. Can this reference be linked to a webpage and last accessed please?

Figure 1

This figure took me some time to decipher and is another way of showing AUCs; perhaps could be clearer in terms of graphics for readers.

6. PLOS authors have the option to publish the peer review history of their article (what does this mean? ). If published, this will include your full peer review and any attached files.

**Do you want your identity to be public for this peer review?** For information about this choice, including consent withdrawal, please see our Privacy Policy .

Reviewer #1: **Yes: ** Professor Fuk Hay Tang

Reviewer #2: **Yes: ** David Lee

---

## [Editor Report · Decision Letter 1]

Radiomics Analysis for the Early Diagnosis of Common Sexually Transmitted Infections and Skin Lesions

PDIG-D-24-00497R1

Dear Mr. Sun,

We are pleased to inform you that your manuscript 'Radiomics Analysis for the Early Diagnosis of Common Sexually Transmitted Infections and Skin Lesions' has been provisionally accepted for publication in PLOS Digital Health.

Best regards,

Hisham Al-Obaidi, PHD

Academic Editor

PLOS Digital Health

**Additional Editor Comments (if provided):**

The authors have adequately addressed all concerns, improved the manuscript's clarity, and enhanced the figures and references. I am pleased to recommend acceptance of this manuscript for publication in PLOS Digital Health.